# Gastric motility and pulmonary function in children with functional abdominal pain disorders and asthma: A pathophysiological study

**Manori Vijaya Kumari**[1]**, Lakmali Amarasiri**[2]**, Shaman Rajindrajith**[3]**, Niranga Manjuri Devanarayana**[4]*

**1** Department of Physiology, Faculty of Medicine & Allied Sciences, Rajarata University of Sri Lanka, Anuradhapura, North Central Province, Sri Lanka, **2** Department of Physiology, Faculty of Medicine, University of Colombo, Colombo, Western Province, Sri Lanka, **3** Department of Pediatrics, Faculty of Medicine, University of Colombo, Colombo, Western Province, Sri Lanka, **4** Department of Physiology, Faculty of Medicine, University of Kelaniya, Ragama, Western Province, Sri Lanka

* niranga@kln.ac.lk

**Data Availability Statement:** All relevant data are within the manuscript and its Supporting Information files. supplementary file with the

## Abstract

### Background

An association has been shown between functional abdominal pain disorders (FAPDs) and asthma. However, the exact reason for this association is obscured. The main objective of this study is to identify the possible underlying pathophysiological mechanisms for the association between FAPDs and asthma using gastric motility and lung function tests.

### Methods

This was a cross-sectional comparative study that consisted of four study groups. Twenty-four children (age 7–12 years) each were recruited for four study groups; asthma only, FAPDs only, both asthma and FAPDs, and healthy controls. Asthma was diagnosed using the history and bronchodilator reversibility test. The diagnosis of FAPDs was made using Rome IV criteria. All subjects underwent ultrasound assessment of gastric motility and pulmonary function assessment by spirometry, using validated techniques.

### Results

All gastric motility parameters, gastric emptying rate, amplitude of antral contraction, and antral motility index, were significantly impaired in children with FAPDs only, children with asthma only, and children with both asthma & FAPDs, compared to controls ($p<0.05$). Pulmonary function parameters indicating airway obstruction ($FEV_1$/FVC ratio, peak expiratory flow rate, FEF25-75%) were not impaired in children with FAPDs only compared to controls ($p>0.05$), but significantly impaired in children with asthma and children with both disorders. Antral motility index correlated with the $FEV_1$/FVC ratio ($r = 0.60$, $p = 0.002$) and FEF25%-75% ($r = 0.49$, $p = 0.01$) in children with both asthma and FAPDs.

manuscript. We confirm that the following are provided in the attached data file. The values behind the means, standard deviations and other measures reported; The values used to build graphs; The points extracted from images for analysis.

**Funding:** This study was carried out with the personal funds of the first author. The authors received no specific funding for this work.

**Competing interests:** The authors have declared that no competing interests exist.

## Conclusions

Gastric motor functions were significantly impaired in children with asthma, children with FAPDs, and children with both disorders. Motility index, measuring overall gastric motor activity, showed a significant positive correlation with lung function parameters that measure airflow limitation. Therefore, these diseases might arise as a result of primary disturbance of smooth muscle activity in the airways and gastrointestinal wall, which could be a possible pathophysiological mechanism for this association between asthma and FAPDs.

## Introduction

Functional abdominal pain disorders (FAPDs) are highly prevalent in children across the world, with an estimated prevalence of 13.5% [1]. High prevalence, healthcare expenditure, and negative impact on quality of life make FAPDs a major global health problem in children [2, 3]. The Rome IV classification identifies four types of FAPDs in children; functional dyspepsia (FD), irritable bowel syndrome (IBS), functional abdominal pain-not otherwise specified (FAP-NOS), and abdominal migraine (AM) [4].

Asthma is a disorder of the small airways characterized by obstruction, hyper-responsiveness, and chronic inflammation. Like FAPDs, with high prevalence ranging from 0.8–32.6% worldwide and poor quality of life, asthma is also considered a significant health problem in children across the world [5, 6].

White *et al.* demonstrated an increased bronchial hyper-responsiveness to inhaled methacholine in patients with IBS speculating the potential association between asthma and FAPDs [7]. Several studies have demonstrated the association between IBS and asthma in children and adults [8, 9]. However, the exact pathophysiological mechanisms that could explain this association have not been elucidated [10]. Therefore, the main objective of this study was to find a possible pathophysiological mechanism to explain this association between asthma and FAPDs.

The pathophysiology of asthma is complex and involves airway hyper-responsiveness, mucosal inflammation, airflow limitation, immunological reactions, smooth muscle dysfunction, and psychological factors [11]. Similarly, visceral hypersensitivity, inflammation of gut mucosa, impaired gastrointestinal motility, and complex interactions between the bidirectional brain-gut axis among the main pathophysiological mechanisms in FAPDs [12, 13].

Both the lung and the upper gut have the same embryological origin from the foregut, and the airways and gastrointestinal tract contain smooth muscle cells [14]. Therefore, in this study, we hypothesized that smooth muscle dysfunction is a shared pathophysiological mechanism for the observed association between FAPDs and asthma. To test our hypothesis, we assessed gastrointestinal smooth muscle function and bronchial smooth muscle function indirectly using standard tests.

## Materials and methods

### Study design and study setting

This was a cross-sectional comparative study. We conducted this study in the Lung Function Laboratory and Gastroenterology Research Laboratory of the Faculty of Medicine, University of Kelaniya, Sri Lanka. The study consisted of four groups; children with only asthma, children

with only FAPDs, children with both diseases (asthma & FAPDs), and healthy children free from gastrointestinal and respiratory symptoms.

## Selection of study participant

Children with asthma and FAPDs were recruited from pediatric outpatient clinics of North Colombo Teaching Hospital, Ragama, Sri Lanka. Healthy children were recruited as controls. Their age ranges from 7–12 years. When recruiting healthy controls, children with a history of chronic medical or surgical disorders other than asthma and FAPDs and those on long-term medication were excluded from the study. Possible organic disorders were ruled out by history, physical examination, and investigations performed in all patients. All recruited patients underwent investigations including complete blood count, urine microscopy, urine culture, and acute-phase proteins. Some children underwent specific investigations depending on the clinical judgment of the consultant pediatrician, included renal function test ($n = 9$), serum amylase ($n = 2$), liver profile ($n = 12$), abdominal ultrasound ($n = 28$), X-ray urinary tract ($n = 1$), and lower gastrointestinal endoscopy ($n = 2$). None of the patients showed clinical or laboratory evidence of organic diseases other than asthma and FAPDs.

## Diagnosis of asthma

Children with a diagnosis of asthma were recruited, and their diagnosis was confirmed by history and the bronchodilator reversibility test following the National Institute for Health and Care Excellence guideline [15]. Both short and long-acting bronchodilators were stopped 6 hours and 12 hours before the reversibility test, respectively. Most of the children with asthma are on inhaled steroids; however, none were on oral steroids.

## Diagnosis of FAPDs

We used Rome IV criteria to diagnose FAPDs in children [4].

## Assessment of gastric motility

Real-time ultrasonography was a previously validated method of assessing gastric emptying and antral motility [16]. A high-resolution real-time scanner (Siemens ACUSON X300 ™) with 1.8MHz to 6.4 MHz curved linear transducer and facilities to record and playback was used to measure gastric motility parameters. Ultrasonographic assessment of gastric emptying is a well-established method of assessing gastric motility, and it shows a clear correlation with the gold standard scintigraphic method [17].

## Gastric emptying rate

After overnight fasting, all subjects were asked to drink a standard liquid meal (200mL of chicken broth, heated to approximately 40˚C, drunk within 2 min). The ultrasound probe was placed over the abdomen vertically to visualize the pyloric antrum, superior mesenteric artery, and abdominal aorta simultaneously. The measurement of the cross-sectional area of the antrum was obtained at fasting state, and 1min and 15min after drinking the liquid meal by tracing the mucosal side of the wall using the built-in caliper. The gastric emptying rate was computed using the following formula [16].

Gastric emptying rate (%) = ([Antral area at 1min–Antral area at 15min]/Antral area at 1min) X100

## Calculation of antral motility

Antral motility parameters were obtained within the first 5 minutes after drinking the liquid meal. The cross-sectional area of the antrum was recorded during contractions and relaxations three times to calculate the amplitude of antral contractions. The antral motility index is a measure of the overall contractile activity of the gastric antrum.

Antral motility parameters were calculated as follows:

- Frequency of antral contractions = Number of contractions per 3 min

- Amplitude (%) = ([Antral area at relaxation–Antral area at contraction]/Antral area at relaxation) X 100

- Antral motility index = Amplitude of antral contractions X Frequency of contractions

## Assessment of lung function

Spirometry was performed using a Vitalograph Alpha Touch spirometer (Vitalograph Ltd. UK) on all subjects, according to National Institute for Healthcare Excellence guideline [15]. The maneuver was first demonstrated to children. The subjects were requested to stand during the procedure. A nose clip was applied to prevent breathing through the nose. They were instructed to expire forcefully and maximally until they feel that no breath is left and then inspire rapidly to maximum capacity. Each maneuver was monitored by flow-volume loops and manual observation to ensure that the effort was maximal, smooth, and artifact-free. The blow with the highest recording was used for the analysis. The above procedure was repeated 15 minutes after inhaling 200 mcg of salbutamol from a metered-dose inhaler via a spacer to assess the bronchodilator reversibility. Main lung function parameters recorded were forced vital capacity (FVC), Forced Expired Volume in the first second ($FEV_1$), Forced Expiratory Flow between 25–75% (FEF25%-75%), Forced Expiratory Flow at 50% (FEF50%), and Peak Expiratory Flow Rate (PEFR).

## Ethical approval

The protocol of this study conforms to the ethical guidelines of the Declaration of Helsinki (originated in 1964 as revised in 2000) as reflected in a prior approval by the institution's human research committee. Ethical Review Committee of the Faculty of Medicine, University of Kelaniya, Sri Lanka, approved this study protocol (ERC reference number P/204/10/2015 Date of approval 09/12/2014). We obtained written, fully informed consent from parents of all children before the commencement of the study.

## Statistical analysis

**Sample size calculation.** The sample size was calculated using the statistical software (Winpepi version—11.65).

1. To compare gastric motility between groups
   Since previous studies assessing liquid gastric motility are not available for children with bronchial asthma, the sample size was calculated using data obtained for gastric emptying in children with FAPDs and healthy controls [18]. According to the available data at a significant level of 0.05 and power of 80%, to show 1 SD difference, the minimal sample size required is 17 in each group.

2. To compare the lung functions between groups
   The sample size was estimated using previous research data (SD value) obtained for lung

functions ($FEV_1$, FEF25-75%) in patients with FGIDs and healthy controls [19]. According to the available data at a significant level of 0.05 and power of 80%, to show 1 SD difference, the minimal sample size required is 15 in each group. To increase the validity of our results, we recruited 24 subjects for each group.

### Data analysis

Gastric motility and lung function parameters between four groups were compared using One-Way ANOVA with Post Hoc Test (Tukey HSD). Lung function parameters and gastric motility parameters were correlated using Spearman Rank Correlation. A $p$-value $<0.05$ was considered statistically significant. PSPP statistical software version 1.0.1 was used in all calculations.

## Results

### Sample characteristics

Table 1 displays the sample characteristics in four study groups: FAPDs only, asthma only, both asthma and FAPDs, and healthy controls.

### Comparison of gastric motility parameters between patients and healthy children

Table 2 displays gastric motility parameters between study groups. Gastric emptying rate, the amplitude of antral contractions, frequency of antral contractions, and antral motility index were significantly lower in patients with only asthma, patients with only FAPDs, and patients with both diseases compared to healthy children. After adjusting for the usage of inhaler drugs, antral motility parameters were still significantly impaired in children with asthma compared to healthy children (odds ratio [OR] 0.28, 95% confidence interval [CI] 0.09–0.84, $p<0.05$). The gastric motility parameters were not significantly different between children with only asthma and only FAPDs.

### Comparison of lung function parameters between patients and controls

Table 3 depicts lung function parameters between study groups. Lung function parameters indicating airway obstruction ($FEV_1$/FVC ratio, FEF25-75%) were significantly lower among

**Table 1. Sample characteristics.**

|  | FAPDs only | Asthma only | Both asthma and FAPDs | Healthy control |
|---|---|---|---|---|
| Age in years | 9.4 (1.6) | 9.6 (1.3) | 9.3 (1.3) | 10 (1.3) |
| *Mean (SD)* |  |  |  |  |
| Gender |  |  |  |  |
| Male *n* (%) | 11 (44%) | 14 (58.3%) | 12 (50%) | 14 (56%) |
| Female *n* (%) | 14 (56%) | 10 (41.7%) | 12 (50%) | 11(44%) |
| FAPDs subtypes |  |  |  |  |
| IBS *n* (%) | 2 (8%) | - | 9 (37.5%) | - |
| FD *n* (%) | 10 (40%) | - | 4 (17%) | - |
| FAP *n* (%) | 8 (32%) | - | 9 (37.5%) | - |
| AM *n* (%) | 5 (20%) | - | 2 (8%) | - |

FAPDs, functional abdominal pain disorders; AM, Abdominal migraine; FAP, Functional abdominal pain disorders; FD, Functional dyspepsia; IBS, Irritable bowel syndrome.

**Table 2. Comparison of gastric motility parameters between study groups.**

| Gastric motility parameters | Asthma only | FAPDs only | Both Asthma and FAPDs | Healthy children |
|---|---|---|---|---|
| | *n = 25* | *n = 25* | *n = 25* | *n = 25* |
| | *Mean (SD)* | *Mean (SD)* | *Mean (SD)* | *Mean (SD)* |
| Fasting antral area ($cm^2$) | 2.9 (1.2) * | 3.5 (1.5) ** | 2.9 (1.3) * | 2.1 (1.0) |
| Gastric emptying rate (%) | 39.7 (13.2) *** | 33.9 (13.3) *** | 32.2 (10.0) ***† | 57.5 (14.5) |
| Amplitude of antral contraction (%) | 42.8 (10.0) *** | 41.0 (8.2) *** | 42.6 (14.3) *** | 63.6 (16.9) |
| Frequency of antral contraction (per 3 min) | 8.9 (1.1) | 8.6 (1.0) ** | 8.6 (0.9) ** | 9.4 (0.9) |
| Antral motility index | 3.9 (1.3) *** | 3.5 (0.8) *** | 3.6 (1.3) *** | 6.0 (1.8) |

***$p<0.0001$

**$p<0.01$

*$p<0.05$ compared to healthy children

†$p<0.05$ compared to children with asthma only.

FAPDs, Functional abdominal pain disorders.

children with only asthma and children with both diseases than healthy children. Lung function parameters were not reduced in children with only FAPDs compared to healthy children. There is no statistically significant difference in spirometric values between children with only asthma and children having both diseases.

## Correlation between lung function parameters and gastric motility parameters

In children with both diseases, the motility index had significant positive correlations with $FEV_1$/FVC ratio (Fig 1) and FEF25-75% (Fig 2). No such correlation was observed with other lung function parameters and in children with only asthma and only FAPDs.

## Discussion

To the best of our knowledge, this is the first pediatric study to assess gastric motility and lung function in children with FAPDs and asthma. We found that children with asthma had a significant impairment of gastric motility compared to healthy children. Furthermore, we have demonstrated a correlation between the antral motility index and lung function parameters,

**Table 3. Comparison of lung function parameters between study groups.**

| Lung function parameters | Asthma only | FAPDs only | Both FAPDs & asthma | Health children |
|---|---|---|---|---|
| | *Mean (SD)* | *Mean (SD)* | *Mean (SD)* | *Mean (SD)* |
| FVC | 1.5 (0.3) | 1.7 (0.4) | 1.5 (0.2) | 1.7 (0.3) |
| $FEV_1$ | 1.3 (0.3) *† | 1.6 (0.4) | 1.3 (0.2) † | 1.5 (0.3) |
| $FEV_1$/FVC ratio | 82.6 (13.6) * | 87.3 (6.8) | 82.7 (7.0) * | 90.4 (5.1) |
| PEFR | 175.8 (60.8) *† | 218.8 (46.4) | 178.4 (49.8) *† | 223.9 (48.9) |
| FEF25-75% | 1.4 (0.5) **†† | 2.1 (0.6) | 1.4 (0.5) **†† | 2.1 (0.6) |
| FEF50% | 1.7 (0.6) **†† | 2.4 (0.7) | 1.7 (0.6) **†† | 2.5 (0.7) |

*$p<0.05$ and

**$p< 0.0001$ compared to controls.

†$P<0.05$ and

††$P<0.01$, compared to FAPDs only.

FAPDs, functional abdominal pain disorders; FVC, forced vital capacity; $FEV_1$, forced expiratory volume in one second; PEFR, peak expiratory flow rate; FEF25%-75%, forced expiratory flow between 25–75% and FEF50%, forced expiratory flow at 50%.

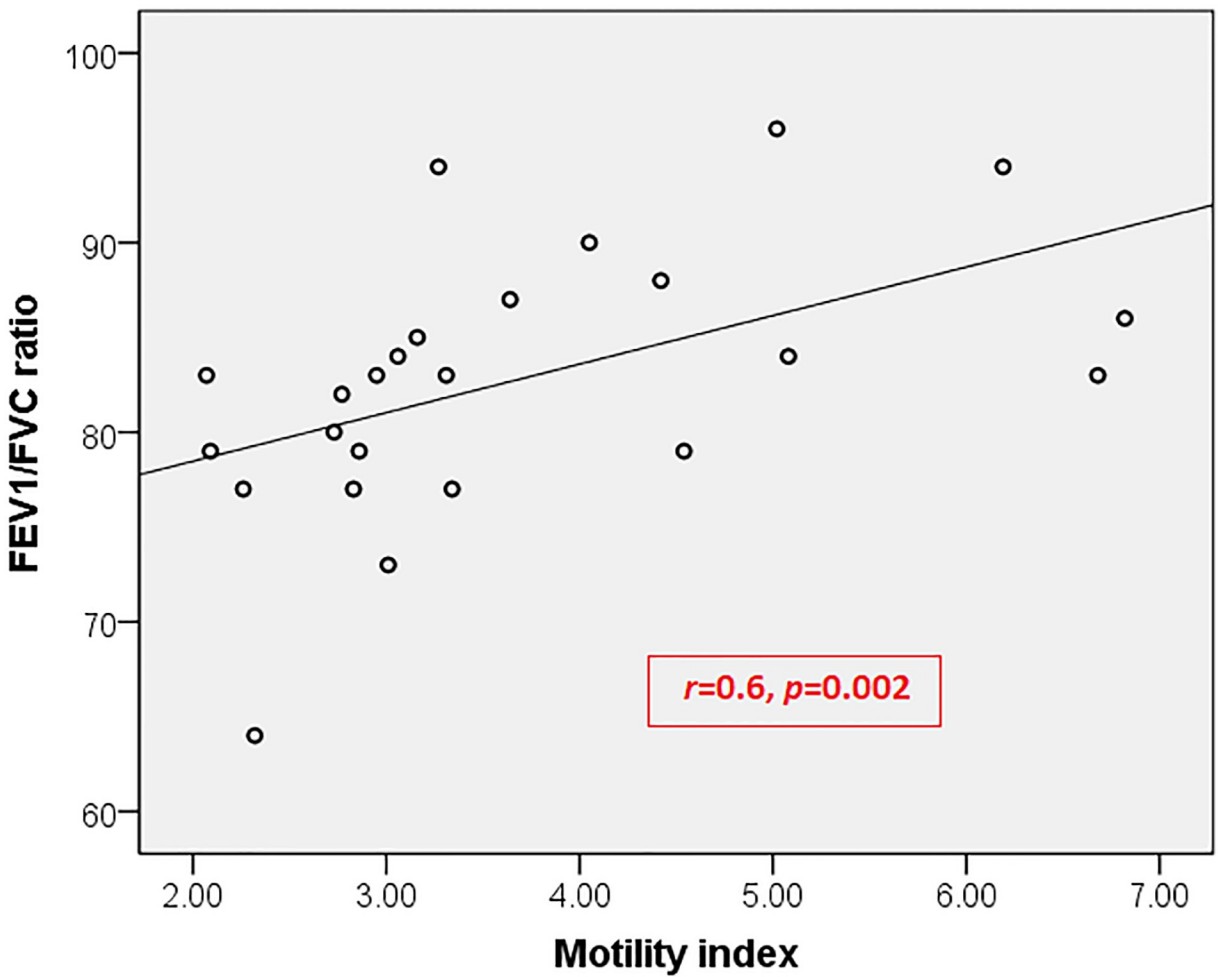

**Fig 1. Correlation between motility index and FEV$_1$/FVC ratio in children with both asthma and functional abdominal pain disorders.** FVC, Forced Vital Capacity; FEV$_1$, Forced expired volume in the first second.

which indicate airway obstruction (FEV$_1$/FVC ratio and FEF25-75%) in children with both disorders. However, there was no significant difference in lung function parameters in children with only FAPDs compared to healthy children.

Several published studies have shown an association between asthma and IBS, mainly in adults [7, 9, 11]. An epidemiological survey, recently reported an association between asthma and different types of FAPDs, namely FAP, FD, and AM, in teenagers aged 13–15 years [8]. However, most studies assessing this association are limited to epidemiological studies, and there is a lack of laboratory-based studies investigating this association in-depth to explain the possible underlying shared pathophysiological mechanisms.

In agreement with our hypothesis, we were able to show gastric dysmotility in children with asthma and FAPDs. This finding indicates that the gastrointestinal dysmotility is common for both diseases and a possible shared pathophysiological mechanism in children with both disorders. Our finding was supported by several previous studies showing a significant impairment of gastric emptying and antral motility in children with FAPDs [18, 20–24]. However, the gastric motor function has never been assessed in children with asthma previously. In accordance

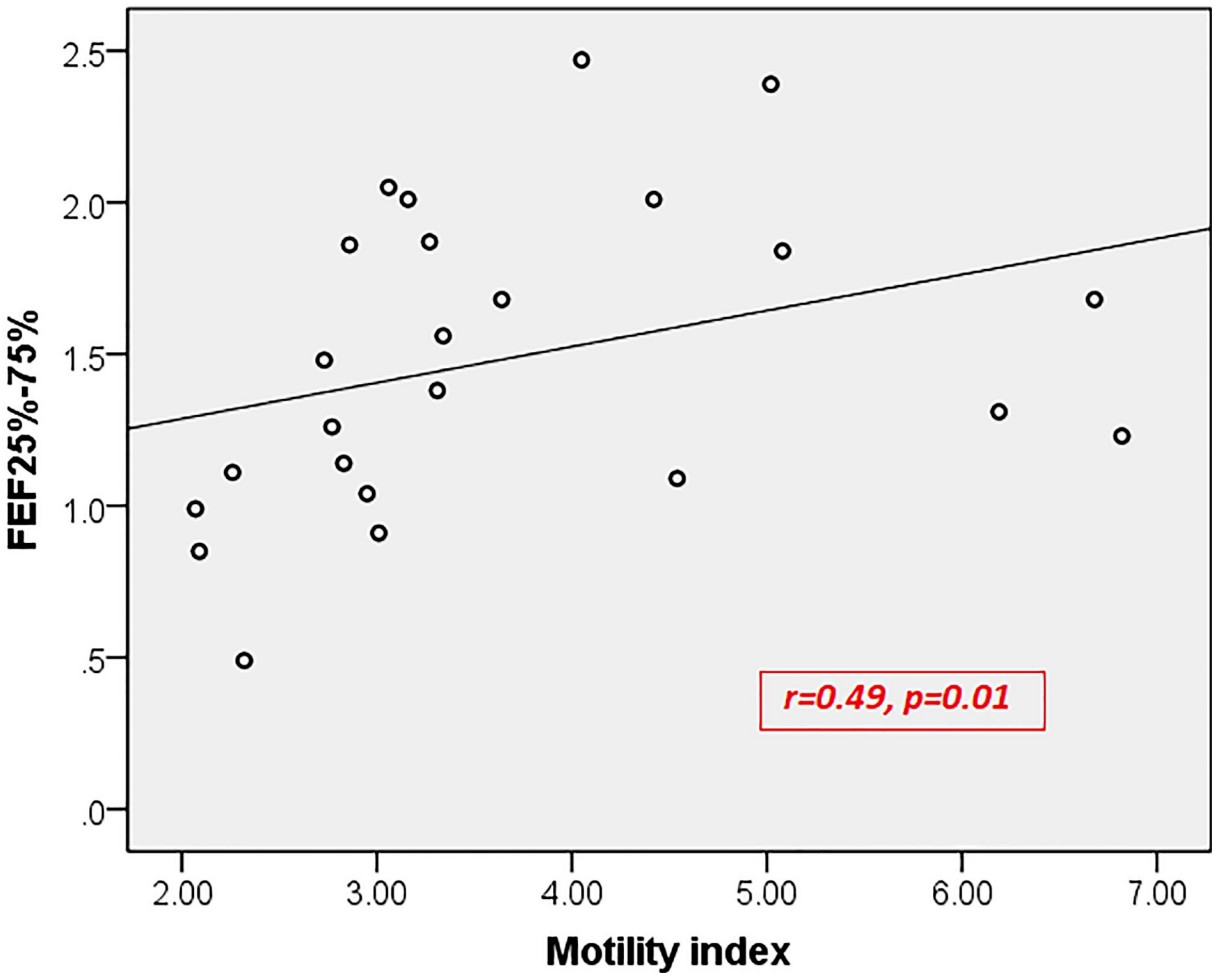

**Fig 2. Correlation between motility index and FEF25-75% in children with both asthma and functional abdominal pain disorders.** FEF25-75%, Forced Expiratory Flow between 25–75%.

with our findings, another study conducted among adults has reported a delayed gastric emptying and lower antral motility index among sufferers of asthma [25].

One possible way of explaining the association between asthma and abnormal gastric function is the presence of gastro-esophageal reflux. Previous studies conducted in both children and adults have shown an association between gastro-esophageal reflux (GER) and asthma. The prevalence of gastro-esophageal reflux disease (GERD) widely varies in patients with asthma. A systematic review, including 19 pediatric studies, reported a prevalence varying from 19.3% to 80.0%. The average prevalence of GER was 22.0% in children with asthma and 4.8% in controls (pooled odds ratio: 5.6 [95% confidence interval: 4.3–6.9]) [26]. This systematic review reported a possible association between GERD and asthma in children but failed to find adequate evidence to support causality. Another study assessing asthma in patients with GERD reported a prevalence of 13.2% (compared to 6.8% in controls) [27]. It is possible that in our cohort of children with FAPDs, increased intragastric pressure due to delayed gastric emptying and impairment of antral motility may predispose them to develop GER precipitating asthma [28]. However, the GER was most likely to be subclinical as none of the patients recruited in the study had clinically overt symptoms of this disease.

In this study, for the first time, we assessed lung functions in children with FAPDs. We could not demonstrate an airflow limitation, indicating airway smooth muscle dysfunction in children with only FAPDs using conventional spirometry. Similarly, two studies conducted among adults failed to detect a significant difference in the spirometry values ($FEV_1$, $FEV_1$/FVC, FEF25-75%) between adult asthmatics with IBS and those without IBS [9, 29]. However, a study in patients with IBS with no respiratory symptoms has demonstrated an increased airway resistance using impulse oscillometry, a more sensitive method than spirometry, indicating the possibility of a subclinical increase in airway resistance and airway smooth muscle dysfunction in patients with IBS [19]. It may be that the abnormal airway resistance in patients with FAPDs is not substantial enough to be detected by direct spirometry, and more sensitive tests, such as body plethysmography and impulse oscillometry, may be needed to detect these changes [30, 31]. Therefore, airway smooth muscle dysfunction could not be completely ruled out as a possible shared pathophysiological mechanism for the association between these two disorders. This possibility is further strengthened by the correlation observed between gastric motility and lung function parameters.

In children suffering from both diseases (asthma and FAPDs), the antral motility index, a measure of the overall contractile activity of gastric antrum, showed significant positive correlations with lung function parameters like $FEV_1$/FVC ratio and FEF25-75%, which are measures of airflow limitation. No such correlation was observed with FVC, which mainly depends on skeletal muscle function. While gastric antral motility is an indirect measurement of the smooth muscle function in the gut, lung function parameters such as $FEV_1$/FVC ratio and FEF25-75% indirectly measure the activity of airway smooth muscles. Therefore, this positive correlation indicates the alteration of smooth muscle activity in the lung and gut among patients suffering from both diseases. On the other hand, it is also possible that delayed gastric emptying and altered antral motility induce subclinical GER, which leads to spasms and dysfunction of the smooth muscles of the respiratory tract, also contributing to the association between FAPDs and asthma as described above. However, we could not detect such a correlation between gastric motility and lung function parameters in children with only asthma and children with only FAPDs. In these two groups, the degree of impairment may not be strong enough to show a definite correlation.

The other possible shared pathophysiological mechanism is immunological dysfunction, common to both disease entities. Increased accumulation of immune-mediated cells, mast cells, eosinophils, and T lymphocytes in the airway mucosa, leads to airway inflammation and airway hyper-responsiveness in children with asthma [32]. Similarly, an accumulation of immune-mediated cells and disordered immune responses were observed in the intestinal mucosa of patients with FAPDs [33, 34]. Mast cells (MCs) are found in intestinal epithelium release a variety of chemicals including serotonin which alters epithelial secretion, intestinal permeability, neuroimmune interactions, visceral sensation, and gastrointestinal movements [35]. MCs found in airway smooth muscle in patients with asthma release immunogenic chemicals, mainly histamine, which shows the dominant effect on contraction of airway smooth muscle cells leading to bronchial hyper-responsiveness and dysmotility in the intestine [36, 37]. Similar to patients with asthma, increased eosinophil count is detected in gastric mucosa in patients with FAPDs and is believed to be responsible for inducing characteristic gastrointestinal symptoms such as pain [38]. Therefore, these findings suggest that the same immunological dysfunctions shared by FAPDs and asthma could be the underlying pathophysiology of this association.

There are several strengths to our study. We have collected data from a sample, which is more than the minimum required sample size, to increase the validity of the results. Furthermore, we have incorporated standard Rome IV criteria in this study to diagnose patients with

FAPDs and pulmonary function tests combined with bronchodilator reversibility to diagnose patients with asthma. We have used standard physiological tests to assess pulmonary functions and gastric motility, which has a good correlation with radio nuclear scintigraphy and excellent reliability and inter-observer variability [17, 39]. One of the limitations of our study was that we recruited patients regardless of their current asthma medication, mainly inhaled corticosteroids. However, the drugs were stopped for an adequate time before laboratory investigations, to minimize their effects on lung function and gastric motility assessments. Furthermore, our findings show that even after adjusting for the usage of inhaled drugs, antral motility parameters were significantly impaired in our asthmatic children compared to healthy children. Another limitation was that we measured smooth muscle dysfunction in the lung and gut indirectly using spirometry and gastric motility ultrasound techniques. We could not use more sensitive measures of airway function such as impulse oscillometry and measures of gastric motility such as scintigraphy due to a lack of resources.

## Conclusion

We report that gastric motor functions are significantly impaired in children with asthma, children with FAPDs, and children with both disorders. Lung function parameters were impaired in children with asthma and children with both diseases but not impaired children with FAPDs. However, the motility index, which measures overall gastric motor activity, showed a significant positive correlation with lung function parameters which measure airflow limitation in children with both diseases. Therefore, it raises the possibility that primary disturbance of smooth muscle activity in the airways and gastrointestinal wall could be a possible shared pathophysiological mechanism for the association between asthma and FAPDs.

## Supporting information

**S1 Data.**
(SAV)

## Acknowledgments

We acknowledge Mrs. J. Ariyawansa and Mr. R.M. Rathnayake, Technical officers, Department of Physiology, Faculty of Medicine, University of Kelaniya, Sri Lanka for their assistance during laboratory procedures.

## Author Contributions

**Conceptualization:** Manori Vijaya Kumari, Shaman Rajindrajith, Niranga Manjuri Devanarayana.

**Data curation:** Manori Vijaya Kumari, Niranga Manjuri Devanarayana.

**Formal analysis:** Manori Vijaya Kumari.

**Funding acquisition:** Manori Vijaya Kumari.

**Investigation:** Manori Vijaya Kumari, Niranga Manjuri Devanarayana.

**Methodology:** Manori Vijaya Kumari, Lakmali Amarasiri, Shaman Rajindrajith, Niranga Manjuri Devanarayana.

**Project administration:** Niranga Manjuri Devanarayana.

**Resources:** Manori Vijaya Kumari.

**Supervision:** Niranga Manjuri Devanarayana.

**Validation:** Manori Vijaya Kumari.

**Writing – original draft:** Manori Vijaya Kumari, Shaman Rajindrajith, Niranga Manjuri Devanarayana.

**Writing – review & editing:** Lakmali Amarasiri.

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
