## [Decision Letter · Decision Letter 0]

20 Apr 2021

PONE-D-21-08562

Gastric motility and pulmonary function in children with functional abdominal pain disorders and asthma: a pathophysiological study

PLOS ONE

Dear Dr. Devanarayana,

Thank you for submitting your manuscript to PLOS ONE. After careful consideration, we feel that it has merit but does not fully meet PLOS ONE’s publication criteria as it currently stands. Therefore, we invite you to submit a revised version of the manuscript that addresses the points raised during the review process.

please correct or write a detailed rebuttal regarding the corrections suggested by reviewers.

We look forward to receiving your revised manuscript.

Kind regards,

Davor Plavec, MD, MSc, PhD, Prof.

Academic Editor

PLOS ONE

Journal Requirements:

Thank you for stating the following financial disclosure:

NO - The funders had no role in study design, data collection and analysis, decision to publish, or preparation of the manuscript.

2a)         Please clarify the sources of funding (financial or material support) for your study. List the grants or organizations that supported your study, including funding received from your institution.

2b)         State what role the funders took in the study. If the funders had no role in your study, please state: “The funders had no role in study design, data collection and analysis, decision to publish, or preparation of the manuscript.”

2c)          If any authors received a salary from any of your funders, please state which authors and which funders.

2d)         If you did not receive any funding for this study, please state: “The authors received no specific funding for this work.”

We note that you have indicated that data from this study are available upon request. PLOS only allows data to be available upon request if there are legal or ethical restrictions on sharing data publicly. For information on unacceptable data access restrictions, please see http://journals.plos.org/plosone/s/data-availability#loc-unacceptable-data-access-restrictions.

3a) If there are ethical or legal restrictions on sharing a de-identified data set, please explain them in detail (e.g., data contain potentially identifying or sensitive patient information) and who has imposed them (e.g., an ethics committee). Please also provide contact information for a data access committee, ethics committee, or other institutional body to which data requests may be sent.

3b) If there are no restrictions, please upload the minimal anonymized data set necessary to replicate your study findings as either Supporting Information files or to a stable, public repository and provide us with the relevant URLs, DOIs, or accession numbers. Please see http://www.bmj.com/content/340/bmj.c181.long for guidelines on how to de-identify and prepare clinical data for publication. For a list of acceptable repositories, please see http://journals.plos.org/plosone/s/data-availability#loc-recommended-repositories.

Additional Editor Comments:

Dear Authors,

please correct or write a detailed rebuttal regarding the corrections suggested by reviewers.

Reviewers' comments:

Reviewer's Responses to Questions

**Comments to the Author**

1. Is the manuscript technically sound, and do the data support the conclusions?

Reviewer #1: Yes

Reviewer #2: Partly

2. Has the statistical analysis been performed appropriately and rigorously? 

Reviewer #1: Yes

Reviewer #2: Yes

3. Have the authors made all data underlying the findings in their manuscript fully available?

Reviewer #1: Yes

Reviewer #2: Yes

4. Is the manuscript presented in an intelligible fashion and written in standard English?

Reviewer #1: Yes

Reviewer #2: No

5. Review Comments to the Author

Reviewer #1: This is a good written paper in which Kumari at al. offer a possible explanation for the previously observed association between functional abdominal pain disorders (FAPDs) and asthma. The authors designed a cross-sectional comparative study to test their hypothesis that smooth muscle dysfunction is one of the possible shared pathophysiological mechanisms in both disorders.

They assessed gastrointestinal and bronchial smooth muscle function in four groups of children: FAPDs only, asthma only, both asthma and FAPDs and healthy controls. The authors performed indirect standard tests: real-time ultrasonography, a previously validated method of investigating gastric emptying and antral motility; and spirometry for assessment of lung function parameters and bronchodilator reversibility. The methods and study design are clearly explained and the study is reproducible. The results were obtained using adequate statistical approach and are presented clearly in the text, tables and figures.

This is the first study investigating gastric motor function in children with asthma and the authors showed that gastric motor functions are significantly impaired in children with asthma, similar to children with FAPDs. This suggests gastrointestinal dysmotility could be a possible shared pathophysiological mechanism between these two disorders.

Several studies investigated spirometry findings in adults with FAPDs, but this is the first paper reporting on lung functions in children with FAPDs. The results show normal lung function test in these children. However, in one subgroup of subjects – children with both asthma and FAPDs, the authors were able to demonstrate a significant correlation between parameters that measure airflow limitation (FEV1/FVC ratio, FEF 25-75%) and the motility index, which measures overall gastric motor activity. These correlations suggest simultaneous alteration of airway and digestive tract smooth muscle activity in patients suffering from both diseases and support the hypothesis of shared pathophysiological mechanisms. However, it is worth mentioning that no such correlation was observed in children with only asthma and only FAPDs.

In the discussion section, the authors address how their findings relate to previous research in this area and offer possible explanations of their results. This section needs revision to improve the understanding and better justify the conclusions, as stated below. The authors correctly identify strengths and limitations of their study. Finally, they conclude by suggesting that both asthma and FAPDs might arise because of primary disturbance of smooth muscle activity in the airways and gastrointestinal wall, which could be a possible pathophysiological mechanism for the association between these disorders.

This study could be a foundation for future research to prove the proposed pathophysiological mechanism with more sensitive measures of airway and gut smooth muscle functions and perhaps lead to development of new effective treatment modalities to manage these children more effectively.

Before approving this paper for publication, I suggest that the authors modify the discussion and clarify the following sections to avoid confusion:

1) Why does more severe impairment of gastric motility in children suffering from both asthma and FAPDs, compared to either disease alone further strengthen the association of gut dysmotility and asthma? (page 18, lines 289-291, and repeated in Conclusion, page 21)

2) How do the authors explain the fact that no correlation was documented between motility index and FEV1/FVC ratio or FEF 25-75% in asthmatic children (Results, page 16, paragraph 1), although the subjects did have impaired gastric motility (especially in the context of a positive correlation found among children with both FAPDs and asthma)? Please add comment to the Discussion section.

3) Is the suggested shared pathophysiological mechanism regarding smooth muscle impairment restricted to children with both diseases, or does subclinical pathology (either respiratory or gastrointestinal) exist in all cases? If yes, what would be the reason for inconsistent correlation results among different study groups?

4) The assertion of the conclusion needs toning down/adjustment accordingly to these revisions. Alternatively, the authors should include more information that clarifies and justifies their composition of the conclusion.

Reviewer #2: Dear author, your research and idea of research are really interesting. Unfortunately, this paper is written in bad English and needs proofreading. Also, I partially disagree with the statement in the conclusion - gastrointestinal dysmotility is a possible common pathophysiological mechanism between these two diseases - as gastrointestinal dysmotility cannot really be a part of asthma pathomechanism... Later, it is well said: therefore, these diseases could arise as a result of a primary disruption of smooth muscle activity in the airways and gastrointestinal wall, which could be a possible pathophysiological mechanism for this association between asthma and FAPDs. I suggest you remove or rephrase the first part.

Also, I noticed that you use the same abbreviation for gastric emptying rate and gastroesophageal reflux, which can be confusing.

6. PLOS authors have the option to publish the peer review history of their article (what does this mean?). If published, this will include your full peer review and any attached files.

Reviewer #1: **Yes: **Duska Tjesic-Drinkovic

Reviewer #2: No

---

## [Author Response · Author response to Decision Letter 0]

21 Jul 2021

Answers to comments of reviewers

Reviewer - 1

Before approving this paper for publication, I suggest that the authors modify the discussion and clarify the following sections to avoid confusion. 

Comment: 

1) Why does more severe impairment of gastric motility in children suffering from both asthma and FAPDs, compared to either disease alone further strengthen the association of gut dysmotility and asthma? (page 18, lines 289-291, and repeated in Conclusion, page 21)

Answer:

Although this statement says there is impairment of gastric motility in children suffering from both asthma and FAPDs compared to either disease alone, it is not actually statistically significant. According to statistical analysis and results (Table 2), gastric motility is impaired significantly in children suffering from both diseases compared to control group (p<0.0001) and compared to children with asthma alone group (p<0.05) but not compared to children with FAPDs alone group (p>0.05). Therefore the above statement was removed from results section, discussion and conclusion of manuscript. 

Comment: 

2) How do the authors explain the fact that no correlation was documented between motility index and FEV1/FVC ratio or FEF 25-75% in asthmatic children (Results, page 16, paragraph 1), although the subjects did have impaired gastric motility (especially in the context of a positive correlation found among children with both FAPDs and asthma)? Please add comment to the Discussion section. 

Answer:

Though it is observed a positive correlation between gastric motility parameters and lung function parameters (FEV1/FVC ratio or FEF 25-75%) in children having both diseases, we could not detect such correlation in children with only asthma and children with only FAPDs. This is because there could be either impairment of airway smooth muscle function at subclinical state in children in children with only FAPDS or or impairment in gastric motility is not severe enough in children with only asthma. Ttherefore, the degree of impairment is not strong enough to give such correlation for them to be associated each other. However, with the diseases progression, at one point, once the smooth muscle dysfunction in lung and gut become severe enough to have a strong correlation, then there is a possibility for these two disorders to get associated with each other sharing the same pathophysiological mechanism. 

Comment: :

3) Is the suggested shared pathophysiological mechanism regarding smooth muscle impairment restricted to children with both diseases, or does subclinical pathology (either respiratory or gastrointestinal) exist in all cases? If yes, what would be the reason for inconsistent correlation results among different study groups?

Answer:

Though a positive correlation is observed between gastric motility and lung function parameters (FEV1/FVC ratio or FEF 25-75%) in children having both diseases, we could not detect such correlation in children with only asthma and children with only FAPDs. This may be because there could be either impairment of airway smooth muscle function or gastric motility function at subclinical state in children with only asthma or in children with only FAPDs, therefore, the degree of impairment is not strong enough to give such correlation for them to be associated each other. However, with the diseases progression, at one point, once the smooth muscle dysfunction in lung and gut become severe enough to have a strong correlation, then there is a possibility for these two disorders to get associated with each other sharing the same pathophysiological mechanism. Therefore, we suggest the possibility that smooth muscle impairment may exist at subclinical state either lung or gut in children with asthma or FAPDs

Comment: 

4) The assertion of the conclusion needs toning down/adjustment accordingly to these revisions. Alternatively, the authors should include more information that clarifies and justifies their composition of the conclusion.

Answer

Revision was made accordingly in the conclusion according to the reviewer comment.

Reviewer – 2

Comment 1

Dear author, your research and idea of research are really interesting. Unfortunately, this paper is written in bad English and needs proofreading.

Answer

English was edited and proofreading was done again

Comment 2

 Also, I partially disagree with the statement in the conclusion - gastrointestinal dysmotility is a possible common pathophysiological mechanism between these two diseases - as gastrointestinal dysmotility cannot really be a part of asthma pathomechanism... Later, it is well said: therefore, these diseases could arise as a result of a primary disruption of smooth muscle activity in the airways and gastrointestinal wall, which could be a possible pathophysiological mechanism for this association between asthma and FAPDs. I suggest you remove or rephrase the first part.

Also, I noticed that you use the same abbreviation for gastric emptying rate and gastroesophageal reflux, which can be confusing.

Answer

According to the reviewer, this statement (gastrointestinal dysmotility is a possible common pathophysiological mechanism between these two diseases) was removed from discussion and conclusion. The abbreviation is also changed.

Editorial Comment

Please amend your current ethics statement to confirm that your named institutional review board or ethics committee specifically approved this study.

Answer

The methods section of the study has been revised according to the comment.

---

## [Decision Letter · Decision Letter 1]

14 Aug 2021

PONE-D-21-08562R1

Gastric motility and pulmonary function in children with functional abdominal pain disorders and asthma: a pathophysiological study

PLOS ONE

Dear Dr. Devanarayana,

Thank you for submitting your manuscript to PLOS ONE. After careful consideration, we feel that it has merit but does not fully meet PLOS ONE’s publication criteria as it currently stands. Therefore, we invite you to submit a revised version of the manuscript that addresses the points raised during the review process.

We look forward to receiving your revised manuscript.

Kind regards,

Davor Plavec, MD, MSc, PhD, Prof.

Academic Editor

PLOS ONE

Journal Requirements:

Additional Editor Comments (if provided):

Please revise the manuscript as suggested by reviewer #1.

Reviewers' comments:

Reviewer's Responses to Questions

**Comments to the Author**

1. If the authors have adequately addressed your comments raised in a previous round of review and you feel that this manuscript is now acceptable for publication, you may indicate that here to bypass the “Comments to the Author” section, enter your conflict of interest statement in the “Confidential to Editor” section, and submit your "Accept" recommendation.

Reviewer #1: (No Response)

Reviewer #2: (No Response)

2. Is the manuscript technically sound, and do the data support the conclusions?

Reviewer #1: Yes

Reviewer #2: (No Response)

3. Has the statistical analysis been performed appropriately and rigorously? 

Reviewer #1: Yes

Reviewer #2: (No Response)

4. Have the authors made all data underlying the findings in their manuscript fully available?

Reviewer #1: Yes

Reviewer #2: (No Response)

5. Is the manuscript presented in an intelligible fashion and written in standard English?

Reviewer #1: Yes

Reviewer #2: (No Response)

6. Review Comments to the Author

Reviewer #1: In the revised manuscript, the authors repeat the statement: “gastric dysmotility is more severe in children having asthma+FAPDs than in having one of those diseases” (Discussion section - line 285), although they acknowledged this is not true (authors’ answer to my Comment 1)

Unfortunately, the authors failed to offer a plausible explanation/answer to my comments 2 and 3. Their conclusions, incorporated in the Discussion section, do not appear sound for several reasons. Specifically, they assume that muscle dysfunction is milder in asthma alone and FAPDs alone than in asthma+FAPDs, but the presented data (Table 2) does not support this assumption. They also suggest that disease progression and the degree of muscle impairment are important for a correlation to exist. They failed to relate to any reference that would back up their point of view. Not being able to find a plausible explanation for a study result is better than offering unsupported or confusing opinions.

Concerning the Reviewer 2 comment – the authors again use the same abbreviation GER for two conditions, what is disappointing. (for example, see lines 290-295).

Therefore, I do not support publishing the work in this form. I encourage the authors to revise the Manuscript once again, mainly the Discussion, following this comment and the previous review comments.

Reviewer #2: (No Response)

7. PLOS authors have the option to publish the peer review history of their article (what does this mean?). If published, this will include your full peer review and any attached files.

Reviewer #1: **Yes: **Duska Tjesic-Drinkovic

Reviewer #2: No

---

## [Author Response · Author response to Decision Letter 1]

21 Sep 2021

Reviewer #1: 

Comment:

In the revised manuscript, the authors repeat the statement: “gastric dysmotility is more severe in children having asthma+FAPDs than in having one of those diseases” (Discussion section - line 285), although they acknowledged this is not true (authors’ answer to my Comment 1)

Answer:

This is revised according to the reviewer comment

Comment:

Unfortunately, the authors failed to offer a plausible explanation/answer to my comments 2 and 3. Their conclusions, incorporated in the Discussion section, do not appear sound for several reasons. Specifically, they assume that muscle dysfunction is milder in asthma alone and FAPDs alone than in asthma+FAPDs, but the presented data (Table 2) does not support this assumption. They also suggest that disease progression and the degree of muscle impairment are important for a correlation to exist. They failed to relate to any reference that would back up their point of view. Not being able to find a plausible explanation for a study result is better than offering unsupported or confusing opinions.

Answer:

This is revised according to the reviewer comment

Concerning the Reviewer 2 comment – the authors again use the same abbreviation GER for two conditions, what is disappointing. (for example, see lines 290-295).

Therefore, I do not support publishing the work in this form. I encourage the authors to revise the Manuscript once again, mainly the Discussion, following this comment and the previous review comments.

Answer:

This is revised according to the reviewer comment

Editorial comments

Comment

We need additional clarification to proceed, please address the following:

1. Please confirm that the minimal data set is shared within your Supporting Information file.

Answer 

We confirm that the following are provided in the attached data file.

The values behind the means, standard deviations and other measures reported;

The values used to build graphs;

The points extracted from images for analysis

---

## [Decision Letter · Decision Letter 2]

17 Dec 2021

Gastric motility and pulmonary function in children with functional abdominal pain disorders and asthma: a pathophysiological study

PONE-D-21-08562R2

Dear Dr. Devanarayana,

We’re pleased to inform you that your manuscript has been judged scientifically suitable for publication and will be formally accepted for publication once it meets all outstanding technical requirements.

Kind regards,

Davor Plavec, MD, MSc, PhD, Prof.

Academic Editor

PLOS ONE

Additional Editor Comments (optional):

Ass all the reviewers suggestions have been accepted the manuscript is acceptable for publication in its current form.

Reviewers' comments:

Reviewer's Responses to Questions

**Comments to the Author**

1. If the authors have adequately addressed your comments raised in a previous round of review and you feel that this manuscript is now acceptable for publication, you may indicate that here to bypass the “Comments to the Author” section, enter your conflict of interest statement in the “Confidential to Editor” section, and submit your "Accept" recommendation.

Reviewer #1: All comments have been addressed

Reviewer #2: (No Response)

2. Is the manuscript technically sound, and do the data support the conclusions?

Reviewer #1: Yes

Reviewer #2: (No Response)

3. Has the statistical analysis been performed appropriately and rigorously? 

Reviewer #1: Yes

Reviewer #2: (No Response)

4. Have the authors made all data underlying the findings in their manuscript fully available?

Reviewer #1: Yes

Reviewer #2: (No Response)

5. Is the manuscript presented in an intelligible fashion and written in standard English?

Reviewer #1: Yes

Reviewer #2: (No Response)

6. Review Comments to the Author

Reviewer #1: (No Response)

Reviewer #2: (No Response)

7. PLOS authors have the option to publish the peer review history of their article (what does this mean?). If published, this will include your full peer review and any attached files.

Reviewer #1: **Yes: **Duska Tjesic-Drinkovic

Reviewer #2: No

---

## [Editor Report · Acceptance letter]

24 Dec 2021

PONE-D-21-08562R2 

Gastric motility and pulmonary function in children with functional abdominal pain disorders and asthma: a pathophysiological study 

Dear Dr. Devanarayana:

I'm pleased to inform you that your manuscript has been deemed suitable for publication in PLOS ONE. Congratulations! Your manuscript is now with our production department. 

Kind regards, 

on behalf of

Dr. Davor Plavec 

Academic Editor

PLOS ONE